# Breeding Dairy Cattle for Female Fertility and Production in the Age of Genomics

**DOI:** 10.3390/vetsci9080434

**Published:** 2022-08-15

**Authors:** Joel Ira Weller, Moran Gershoni, Ephraim Ezra

**Affiliations:** 1Israel Cattle Breeders Association, Caesarea 38900, Israel; 2Agricultural Research Organization, The Volcani Center, Rishon LeZion 15159, Israel

**Keywords:** dairy cattle, female fertility, genomics

## Abstract

**Simple Summary:**

Long-term selection should lead to reduction in heritability, due to fixation of positive alleles. Selection on multiple traits should lead to negative genetic correlations because alleles with positive effects on both traits will respond first to selection, leaving alleles with opposing effects. Efficient selection for milk production traits began in Israel in the 1980s, and selection for female fertility in 2000. Introduction of genomic selection increased rates of genetic gain. Many studies have shown the negative relationships between milk production traits and female fertility. Phenotypic and genetic changes for female fertility and production traits in the Israeli dairy cattle population over the last three decades were studied in order to determine if long-term selection has resulted in reduced heritability and increased negative genetic correlations. Heritabilities were 0.4 for protein production and 0.05 for conception status. The genetic correlation between conception status and protein yield was −0.38. Heritabilities decreased with increase in parity for protein but remained the same for conception status. For milk, fat, and protein production and female fertility, heritabilities increased or stayed the same over the entire period of 30 years. There is no indication that a selection plateau is imminent for dairy cattle.

**Abstract:**

Phenotypic and genetic changes for female fertility and production traits in the Israeli Holstein population over the last three decades were studied in order to determine if long term selection has resulted in reduced heritability and negative genetic correlations. Annual means for conception status, defined as the inverse of the number of inseminations to conception in percent, decreased from 55.6 for cows born in 1983 to 46.5 for cows born in 2018. Mean estimated breeding values increased by 1.8% for cow born in 1981 to cows born in 2018. Phenotypic records from 1988 to 2016 for the nine Israeli breeding index traits were divided into three time periods for multi-trait REML analysis by the individual animal model. For all traits, heritabilities increased or stayed the same for the later time periods. Heritability for conception status was 0.05. The first parity genetic correlation between conception status and protein yield was −0.38. Heritabilities decreased with the increase in parity for protein but remained the same for conception status. Realized genetic trends were greater than expected for cows born from 2008 through 2016 for somatic cell score, conception status and herd-life, and lower than expected for the production traits.

## 1. Introduction

In 1960 Falconer [1] noted that selection on a multiple trait index should lead to the generation of negative genetic correlations among the index traits. This will occur because those polymorphic genes for which the same alleles have positive effects on both traits will respond first to selection, leaving the genes with economically negative correlations. It is also conventional wisdom that selection should result in a reduction in genetic variance, as alleles with positive effects reach fixation. However, as shown by [2] by simulation, and by [3] based on theoretical considerations, in some cases selection can lead to temporary increases in genetic variance due to the increase in frequency of positive rare alleles.

Genomic selection based on genotypes for thousands of genetic markers has been incorporated into nearly all commercial dairy cattle breeding programs over the last decade [4]. In all cases, genomic evaluation is performed on each individual trait included in the selection index without consideration of possible genetic correlations [4].

Female fertility traits were first included in national breeding indexes by the Scandinavian countries in the 1970s. Female fertility was included in the total merit index for Norwegian dairy cattle since 1972 by considering the 56-d nonreturn rate in virgin heifers, that is, the fraction of cows with a first insemination that were not re-inseminated [5]. Israel incorporated “conception status” in the national index in 2000 [6], and the US included “daughter pregnancy rate” in 2003 [7]. Development of the Israeli breeding index since 1985 is given in Table 1. Since 1990, protein production has been given the greatest emphasis in the breeding index. This has been the case for nearly all dairy commercial breeding programs since 2000 [8]. Genomic selection began in Israel in 2015 [9].

Much has been written about both the phenotypic and genetic economically negative correlation between milk production traits and female fertility, reviewed by [10,11]. Numerous studies have attempted to estimate the negative genetic correlations between milk production traits and various measures of female fertility, reviewed by [12].

One major problem with these studies, as noted by [13], is that nearly all measures of female fertility are biased with respect to milk production traits. With respect to the non-return rate, the fact that the cow was not re-inseminated does not mean that the first insemination was successful. The cow could have died, or the farmer could have decided to cull, and farmers are more likely to cull low-producing cows. Similarly, other measures of fertility, such as the number of inseminations per lactation and days open will also be biased, because low production cows are more likely to be culled before a days-open record is generated, and cows with the lowest fertility do not become pregnant. Most studies that have attempted to estimate genetic correlations between fertility and production traits have used sire models and have considered only first parity cows [12].

Insemination data in Israel is unique in that nearly all cows that are inseminated are vet-checked for pregnancy after 60 days [13]. Routine genetic evaluations are computed in Israel for “conception status” (CS), calculated as the inverse of the number of inseminations to conception in percent. For cows that are culled prior to conception or cows for which conception has not yet been recorded, the expected number of inseminations to conception is estimated [14]. Thus, all cows that are inseminated at least once have a record for this trait, and biases with respect to production traits are likely to be minimal.

In order to determine if long-term selection for fertility and production has in fact resulted in reduced heritability and more negative genetic correlations, we estimated the phenotypic and genetic changes for CS over the last three decades, and the environmental and genetic correlations among the traits included in the Israeli breeding index by the individual animal model, including CS and milk, fat, and protein production. Genetic and environmental variance components were computed for first parity for the nine index traits over the last 30 years, and for protein and CS for the first three parities over the last eight years. Genetic trends over the last decade were computed for all traits included in the Israeli breeding index, and the realized values were compared to the expected values derived from the principles of selection index. Finally, genome-wide association studies (GWAS) were performed for both protein production and CS, and correlations were computed between the marker effects.

## 2. Materials and Methods

### 2.1. The Data Sets and Traits Analyzed

The seven data sets analyzed are described in Table 2. The first data set included phenotypic records of cows born from 1988 through 2016 with valid records for herd-life and valid first parity records for the other eight index traits listed in Table 1. For cows born prior to 1988, some of the index traits were recorded only on subsets of the data. Records for cows born after 2016 were incomplete, especially for herd-life. This data set was divided into three parts for estimation of variance components and genetic parameters by restricted maximum likelihood (REML) methodology, as described in Table 2.

The second data set included phenotypic records of cows born between 2008 and 2015 with valid first parity records for protein and CS. This data set was used to estimate variance components and genetic parameters by REML for protein and CS for parities 1 through 3. Genetic and environmental correlations were computed among the three parities for both traits. Both data sets were analyzed by the multi-trait individual animal model (IAM).

The third and fourth data sets were used to compute genetic evaluations for the three milk production traits and CS, respectively, for all cows with valid records born from 1983 through 2018 by the multi-trait IAM, including valid records for parities 1 through 5. Data set 3 included only parities with valid records for all three milk production traits. Data set 5 included estimated breeding values (EBV) for the nine index traits from cows born between 2009 and 2018 and was used to compute realized genetic trends in the Israeli Holstein population. Data sets 6 and 7 included bulls with reliabilities >0.5 for protein and CS respectively, from the analyses of data sets 3 and 4, and genotypes for >40,000 SNPs. These two data sets were used for the GWAS analyses.

Edits applied for all traits included in PD19 prior to genetic evaluation were as described previously [15,16]. All the traits included in PD19, except for herd-life, were analyzed by a multi-trait IAM, with each parity considered a separate trait. Except for the calving traits, all valid parities up to fifth were included in the analyses. In addition to the additive genetic effects, the analysis models included the effects of herd-year-season and parity. The single parity evaluations were then combined into a multi-parity index as described previously [15]. Herd-life was computed as the number of days from first calving to culling and analyzed by a single trait animal model. For cows that have not yet been culled, expected herd-life was computed as described previously [17]. First and second parity dystocia and rate of stillbirth were analyzed jointly by a multi-trait IAM including the effects of the cow calving and the sire of calf as described by [18]. Reliabilities for all traits were estimated as described previously [18,19].

### 2.2. Statistical Analyses

Variance components were estimated for data set 1 with the MTC program [20] and the AIREMLf90 program for data set 2 [21]. The AIREML90 could not be applied to data set 1, in which nine traits were analyzed, because of software limitations. AIREML90 can accept different numbers of records per trait, differing analysis models for each trait and estimates of standard errors for estimates of all variance components and genetic parameters. Both data sets were analyzed by the multi-trait IAM. In addition to the random residual and the random additive genetic effect, data sets 1 and 2 included fixed herd-year-season (HYS) effects. Two seasons were defined for each herd-year, based on freshening month, beginning in April and October of each year. In the analysis of data set 1, the same HYS classes were defined for all nine traits relative to first parity. In the analysis of data set 2, different HYS classes were defined for each parity. In the analysis of data set 1 the same number of records were analyzed for each trait, as required by the MTC program. Therefore, cows with missing records for any of the 9 index traits were deleted. In the analysis of data set 2, the numbers of valid records decreased with an increase in parity. To avoid possible bias, later parity records were included only if there were valid records for the previous parities. In both data sets, all known parents and grandparents of cows with first parity records and of sires of cows with records were included to construct the relationship matrix among animals. The total numbers of animals included in each analysis are also given in Table 2. For both data sets, two genetic groups were defined for animals with unknown parents or animals of the first generation in the data set, one for males and one for females.

Data sets 3 and 4, which included all valid records for cows born between 1983 and 2018, were analyzed by the multi-trait IAM. These models included fixed parity and HYS effects, as defined previously, in addition to the random additive genetic and residual effects. Records were pre-adjusted for birth and calving month and days open as described previously [15,16]. The magnitudes of the residual and genetic variance components were assumed to be known. A total of 92 groups for the milk production traits and 80 groups for CS were defined, based on the sex of the animal with unknown parents, which parent was unknown, and the birth year. In addition, separate groups were defined for sire of cows of breeds other than Holstein. Although <2% of the cows were sired by bulls of other breeds, these bulls were a significant fraction of the total number of bulls, and an even larger fraction of the bulls with unknown parents. The genetic base of the evaluations for all traits was set as the mean EBV of calves born in 2015.

The contribution of each of the nine traits included in the PD19 selection index (*c_j_*) was computed as proposed by [22]:(1)$$c_{j}=\frac{abs(b_{j}g_{j})}{\sum_{j=1}^{J}abs(b_{j}g_{j})}$$
where *b_j_* = the index coefficient for trait *j*, *g_j_* = the genetic standard deviation for trait *j*, *J* = 9, the total number of traits, and “*abs*” denotes “absolute value”. The values of *b_j_* were derived from Table 1, and the values for *g_j_* were derived from the REML analysis of data set 1 for cows born between 2008 and 2016.

Realized genetic gains for the nine index traits were estimated from data set 5 as the linear regressions of the cows’ EBV for each trait on their birth dates. The vector of expected genetic changes over 10 years of selection on PD19 (**Φ**) were computed using the following equation [23]:**Φ** = i**bG**/(**b′Pb**)^0.5^(2)
where i = the selection intensity, **b** = the vector of breeding index coefficients for PD19, **G** = the genetic variance matrix among the 9 traits, and **P** = the phenotypic variance matrix, computed as the sum of the genetic and residual variance matrices. The **G** and **P** matrices were derived from the REML analysis of data set 1 for cows born between 2008 and 2016. The index coefficients for the Israeli breeding index are given in Table 1. The expected economic gain for PD19, TEG, was computed as follows:TEG = **Φ′b**(3)
with all terms as defined previously. The realized gain for PD19 was computed in the same manner with **Φ** replaced by the vector of realized gains, as derived from the analysis of data set 5. The selection intensity in equation [2] is a function of the selection intensities along the four paths of selection and is only known approximately. Therefore, i was set to 3.02 so that the expected economic gain for PD19 should be equal to the realized gain.

### 2.3. Genomic Analysis

The genomic analysis included all Israeli Holstein bulls with genotypes born from 1991 with reliabilities >0.5 from the analyses of data sets 3 and 4. Of the 1749 Israeli Holstein bulls genotyped, 1663 were born since 1991 and had genetic evaluations for protein with reliabilities >0.5, and 1610 had genetic evaluations for CS. As genotyping of these bulls was performed using several SNP chip platforms, we included only those markers that were genotyped in >90% of the tested cohort. A total of 40,498 SNPs were retained. GWAS files were prepared and formatted as described [24,25] using the plink software [26]. The allele substitution effects and the nominal probabilities for the hypothesis of no effect were computed using EMMAX software [27]. To account for the effects of relationships among individuals, we generated a pseudo-relationship matrix based on the identity by state matrix calculated using the emmax-kin-intel64 algorithm and the -v -s -d 10 flags. The GWAS was computed using the EMMAX algorithm with the -v -d 10 -t flags, and the relationship matrix was included in the analysis using the -k argument. Experiment-wise probabilities accounting for multiple testing were computed based on the Bonferroni correction. To assess the variance explained by all SNPs, we used the kinship matrix and the bull phenotypes list and calculated the EMMAX software genomic REML that provides pseudo-heritability estimates [27].

## 3. Results

First parity phenotypic annual means for CS for cows born from 1983 through 2018 and annual means of EBV for CS of cows born from 1981 through 2018 are given in Figure 1. In correspondence with the results presented by [10], annual phenotypic means for CS decreased from 55.6% for cows born in 1983 to 46.5% for cows born in 2018, while mean EBV for CS increased from −1.9 for cow born in 1981 to −0.1 for cows born in 2018, the last year with nearly complete data. As noted previously, the genetic base was set to the mean of cows born in 2015. CS was added to the Israeli breeding index in 2000 (Table 1). Mean annual EBV increased by 2.9% from 2000 to 2018.

Annual genetic means for the EBV of the three milk production traits are given in Figure 2. The slopes for fat and protein EBV were nearly linear during the entire period from 1982 through 2018. The slope for milk is higher in the early years and declines after 1992. This corresponds to the major changes in the breeding index in 1990 and 1991 (Table 1). Genetic gains were nearly equal for fat and protein, despite the fact that the index coefficient for protein was more than double the coefficient for fat since 1991. This occurs because both the heritability and genetic variance for fat are greater than for protein.

Genetic parameters derived from the REML analysis of data set 1 are given in Table 3. Heritabilities were generally highest for milk, followed by fat and protein. Heritability for CS was between 0.04 and 0.05 for the three time periods. For all traits, heritabilities increased or stayed the same for the later time periods, despite the “conventional wisdom” that long-term selection should reduce heritability. Genetic correlations between the 3 milk production traits and CS were all negative, and the absolute value was highest for the correlation between CS and protein, between −0.37 and −0.38. Similar to the heritabilities, genetic correlations did not decrease over time. Environmental correlations between the three milk production traits and CS were very close to zero, despite previous studies that indicate a negative phenotypic relationship between milk production traits and female fertility [9].

Genetic parameters for protein and CS ± standard errors in the first three parities as derived from the REML analysis of data set 2 are given in Table 4. Since this data set was analyzed by the AIREMLf90 program, approximate standard errors were computed. All standard errors of the heritabilities and genetic and environmental correlations were <0.05. First parity heritabilities for protein and CS was nearly equal to the first parity heritabilities in Table 3 for the most recent time period. For both traits, heritabilities decreased with increase in parity, but proportionally less for CS. Genetic correlations among parities were all <0.65 for protein, and >0.89 for CS. All genetic correlations between protein and CS were negative, but the absolute value of −0.34 was highest for the first parity. This absolute value is slightly lower than the absolute value of −0.38 in the analysis of data set 1, but not significantly different. The absolute values of genetic correlations between protein and CS decrease with the increase of parity to −0.12 for the third parity. Environmental correlations between protein and CS for the three parities were positive, but all were <0.07, as compared to nearly zero in data set 1.

Genetic and environmental variance components from the analysis of data set 1 for cows born from 2008–2016 are given in Table 5. These values are similar to previous analyses of the Israeli dairy cattle population [28].

Genetic standard deviations (SD), fraction of the Israeli breeding index, PD19 as computed by equation [1], the expected genetic trends, as computed by equation [2], and realized genetic trends, as computed by the regression of EBV on the cows’ birth dates for cows born from 2008 through 2016, are given in Table 6. The genetic SD and the genetic and phenotypic variances in equation [2] were derived from the genetic and environmental variances in Table 5. The difference between the realized and expected trends, divided by the genetic SD of each trait, is also given. For the traits in which negative values are economically favorable, the sign of realized-expected is reversed so that a positive value indicates that the realized genetic gain in the desired direction was greater than the expected gain. Realized trends were greater than expected for SCS, CS and herd-life, with the largest difference for herd-life. This is not too surprising, since natural selection also favors increased herd-life and fertility, and lower mastitis. Farmers also prefer high fertility bulls. In addition, the expected trends were based only on first parity records, while the realized trends were based on all parities up to the fifth. As shown in Table 4, heritabilities decreased with an increase in parity proportionately more for protein, as compared to CS, and genetic correlations among parities were higher for CS. Both factors should result in increased realized genetic gain for CS, as compared to protein. The absolute values of the discrepancy between the realized and expected genetic trends were lowest for protein, the trait with the largest fraction of the index, and the highest for milk. The large discrepancy for milk is probably due to the fact that farmers prefer bulls with high fat and protein concentrations, even though the concentration traits are not included in the index. Production quotas are in fluid milk, while payment is for fat and protein. Thus, farmers can increase total income using high concentration bulls.

The EMMAX G-REML results for CS and protein were 0.63 and 0.87. That is, the cumulative effects of all the markers explained 63% and 87% of the variance among the sire evaluations for these traits. After the Bonferroni correction for multiple comparisons, none of the markers for CS were significant at the 5% experiment-wise level, and only three markers met this criterion for protein.

The scatter plot of the allele substitution effects of 31,744 SNPs with minor allele frequencies (MAF) >0.10 on protein as a function of their effects on CS is given in Figure 3. As can be seen on the level of all markers, there was virtually no relationship between the marker effects on the two traits. The coefficient of determination was <10^−4^. The scatter plot of the 10 markers with the lowest probability values for protein as a function of the effects of these markers on CS is given in Figure 4. For these markers the coefficient of determination was 0.01, that is a correlation of ~0.1, and the slope was not significantly different from zero. Thus, the overall genetic correlation between these traits was not reflected by the substitution effects of the individual markers.

## 4. Discussion

“Cow conception rate” in the US dairy cattle population is defined as the percentage of inseminated cows that become pregnant at each service, and “daughter pregnancy rate” is defined as the percentage of nonpregnant cows that become pregnant during each 21-day period [29]. Heritabilities for the daughter pregnancy rate and cow conception rate were estimated at 0.04 and 0.02 in 2014, and the genetic correlation between the two traits was estimated as 0.87. Genetic correlations with protein production were −0.18 and −0.15 [30], but details of the analysis model were not given. CS is apparently closer to the cow conception rate and has similar heritability. The number of services, pregnancy/conception to first service, and pregnancy within a given period were included among the fertility traits considered in the meta-analysis of [11]. Mean heritabilities for these traits among the studies analyzed ranged from 0.02 to 0.03. Thus, heritability of CS in the current study is at the high end for traits that estimate the probability of cow conception. The mean of genetic correlations of protein production with the three traits considered by [11] were 0.35, −0.37 and −0.17. Thus, correlations of protein with number of services (which was positive, because a high value indicates low fertility) and pregnancy/conception to first service were very similar to the correlation between protein and CS in first parity, despite the fact that CS should be less biased by the culling of cows with low production [13,14].

There is an apparent contradiction between the major phenotypic reduction in CS over the last 30 years, and the fact that the environmental correlations between CS and the milk production traits were all very low. It should be noted, however, that the HYS effect was considered fixed in the variance component analyses. Thus, changes in management factors detrimental to fertility would be excluded from the environmental correlations.

The genetic trend for CS in the Israeli Holstein population became positive in 2001, the year after this trait was included in the Israeli breeding index. Genetic trends in the UK and Ireland for calving interval were negative until 2004 and then positive since then, corresponding to the changes in the selection indices of these countries [11]. Genetic evaluations for the DPR of US Holsteins first became available to the industry in 2003. The genetic trend for DPR was negative until 2009 and has since been positive [31]. Thus, it is clear from the results for all four countries that positive selection for female fertility is possible without a major decrease in selection for production traits.

In the current study, changes in heritability and genetic correlations were investigated for close to 30 years, or approximately six generations. In long-term selection experiments, the selection response usually ends after 20 to 30 generations [32], although, in some cases, a significant response has continued for over 100 generations [33]. Despite these considerations, the analysis of several studies indicates that heritability of lactation milk yield in dairy cattle has risen from ~25% in the 1950s to ~35% in the past decade, although this may be largely due to improved management [34]. To the best of our knowledge, this is the first study that attempted to estimate changes in heritabilities and genetic correlations over time for economic traits in dairy cattle in a specific population. The slight increase in heritabilities corresponds to the results of [34]. Thus, as noted previously by [3], there is no reason to assume that these traits are approaching a selection plateau.

Current methods applied for genomic evaluation in nearly all populations are based on the evaluation of bull EBV separately for each trait [4]. These methods cannot be readily adapted to multi-trait genomic evaluation. The method of [35] is based on the analysis of the phenotypic records with the inclusion of the genomic relationship matrix, as derived from the marker genotypes. It should be theoretically possible to apply this method to multi-trait genomic evaluation. Results of the GWAS demonstrate that the overall negative genetic correlation between protein production and female fertility is not reflected in the effects of the individual markers.

## 5. Conclusions

Annual means for CS decreased from 55.6 for cows born in 1983 to 46.5 for cows born in 2018, while mean estimated breeding values increased from −1.9 for cows born in 1981 to −0.1 for cows born in 2018. Heritability for CS increased from 0.04 to 0.05 from the first through the third time periods. For all traits, heritabilities increased or stayed the same for the later time periods. The first parity genetic correlation between conception status and protein was −0.38. Environmental correlations between CS and the milk production traits were all close to zero. Heritabilities decreased with an increase in parity for protein but remained the same for CS. Realized trends were greater than expected for cows born from 2008 through 2016 for SCS, CS and herd-life, and were lower than expected for the production traits. No correlation was found for the effects on protein and conception status of the 31,744 SNPs with minor allele frequency >0.1 included in the GWAS analysis, or for the 10 markers with the lowest probability values for the substitution effect on protein. Thus, there is no apparent reason to change the current procedures that compute genomic evaluations separately for each trait.

## Figures and Tables

**Figure 1 vetsci-09-00434-f001:**
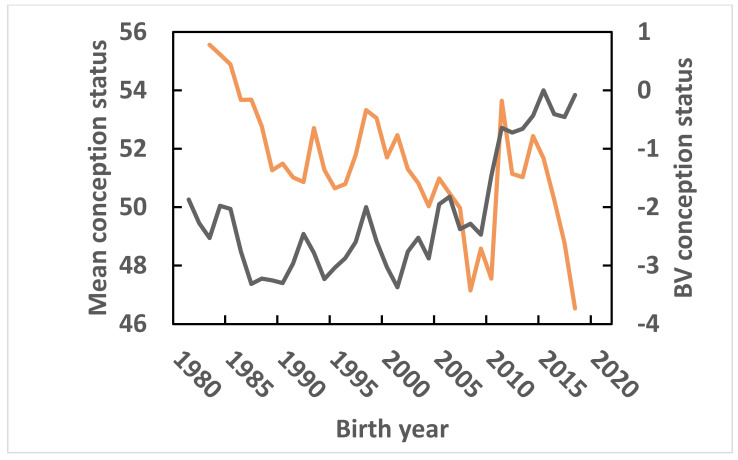
First parity phenotypic annual means for conception status, defined as the inverse of the number of inseminations to conception in percent, for cows born since 1983, orange line; and annual means of estimated breeding values (EBV) for conception status of cows born since 1981, black line. The genetic base was set to cows born in 2015.

**Figure 2 vetsci-09-00434-f002:**
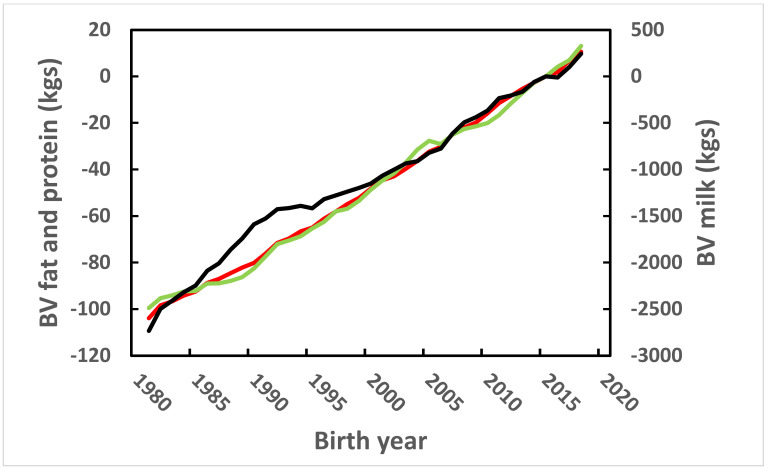
Annual estimated breeding values (EBV) means of cows born since 1982 for milk, black line; fat, green line; and protein, red line. The genetic bases for all traits were set to cows born in 2015.

**Figure 3 vetsci-09-00434-f003:**
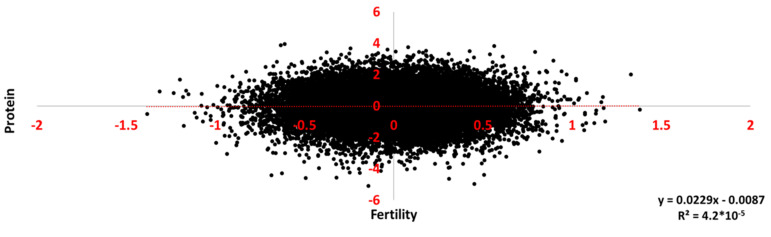
Allele substitution effects on kg protein of all SNPs with minor allele frequencies >10%, as a function of their substitution effects on female fertility, defined as the inverse of the number of inseminations to conception in percent.

**Figure 4 vetsci-09-00434-f004:**
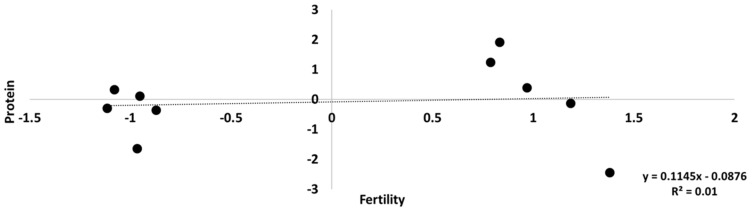
Allele substitution effects of the 10 markers with the lowest probability values for protein as a function of the effects of these markers on female fertility, defined as the inverse of the number of inseminations to conception in percent.

**Table 1 vetsci-09-00434-t001:** Development of the Israeli breeding index since 1985. Table values are index coefficients.

	Year of Change										
Traits	1985	1990	1991	1996	2000	2001	2004	2007	2011	2016	2019
Milk (kg)	0.51	0	−0.274	−0.274	−0.274	−0.22	0	0	0	0	0
Fat (kg)	14	0	6.41	6.41	6.41	8.5	6.3	6.3	7.9	8.48	9.94
Protein (kg)		1.0	34.85	34.85	34.85	31.0	25.4	25.4	23.7	21.2	19.88
SCS ^1,2^				−300	−300	−300	−300	−300	−300	−300	−300
CS ^3^ (%)					26	26	26	26	26	26	26
Herd-life (days)							0.6	0.6	0.6	0.6	0.6
Persistency (%)								10	10	10	10
Dystocia (%) ^2^								−3	−3	−3	−3
Stillbirth (%) ^2^								−6	−6	−6	−6

^1^ Somatic cell score. ^2^ Negative values are economically favorable. ^3^ Conception status, defined in the text.

**Table 2 vetsci-09-00434-t002:** Basic description of the seven data sets analyzed.

Data	Trait	Analysis	Parities	Birth Years		Number of			
Set	Analyzed	Type ^1^		Beginning	End	Records	HYS	Animals	Genetic Groups
1	Phenotypic	MVC	1	1988	1997	177,073	14,715	290,921	2
	values for the		1	1998	2007	213,495	15,169	349,012	2
	9 index traits ^2^		1	2008	2016	234,276	11,041	377,631	2
2	Phenotypic	MVC ^4^	1	2008	2015	229,036	10,634	370,999	2
	values for		2			186,687	10,558		
	Protein and CS ^3^		3			129,576	9887		
3	Phenotypic values for milk production traits	MAM	1–5	1983	2018	1,070,284	50,630	1,198,095	92
4	Phenotypic values for CS	MAM	1–5	1983	2018	970,883	54,396	1,126,063	80
5	EBV of cows for 9 index traits	Regression ^5^	1–5	2009	2018	359,202	-	359,202	-
6	EBV of bulls for protein	GWAS ^6^	1–5	1991	2016	1663	-	1663	-
7	EBV of bulls for CS	GWAS	1–5	1991	2016	1610	-	1610	-

^1^ MVC = Multi-trait REML variance component analysis based on the animal model, MAM = multi-trait animal model with variance components assumed known, GWAS = genome-wide association study, based on the genetic evaluations of sires. ^2^ Index traits for PD19 are given in Table 1. ^3^ Conception status, see text for details. ^4^ A single multi-trait analysis was performed including parities 1–3 of protein and CS. ^5^ Regression of the cows’ breeding values on their birth years. ^6^ Birth years and number of records refer to bulls with reliabilities >0.5.

**Table 3 vetsci-09-00434-t003:** Genetic parameters derived from the REML analysis of data set 1. Heritabilities on the diagonal in **bold** type. Genetic correlations above the diagonal and environmental correlations below the diagonal.

Birth Years	Traits	Milk	Fat	Protein	SCS	CS	Herd-Life	Persistency	DC	SB
1988–	Milk	**0.47**	0.41	0.71	0.13	−0.34	0.12	0.10	−0.04	0.00
1997	Fat	0.55	**0.47**	0.56	0.04	−0.25	0.17	0.05	0.10	0.09
	Protein	0.84	0.63	**0.41**	0.17	−0.38	0.14	0.05	0.02	0.05
	SCS	−0.02	−0.02	0.02	**0.23**	−0.21	−0.33	−0.06	0.00	0.05
	CS	0.01	−0.01	0.00	−0.01	**0.04**	0.41	0.13	−0.30	−0.28
	HL	0.14	0.11	0.14	−0.09	0.12	**0.10**	0.29	−0.15	−0.13
	Persistency	0.01	0.00	−0.01	−0.04	−0.02	0.08	**0.18**	−0.06	−0.03
	Dystocia	−0.03	−0.01	−0.03	−0.01	−0.04	0.03	0.01	**0.04**	0.91
	Stillbirth	−0.03	−0.02	−0.02	−0.01	−0.03	0.00	0.00	0.41	**0.03**
1998–	Milk	**0.49**	0.40	0.77	0.14	−0.30	0.03	0.21	−0.05	−0.02
2007	Fat	0.56	**0.46**	0.55	0.04	−0.22	0.05	0.04	0.00	−0.01
	Protein	0.87	0.65	**0.41**	0.16	−0.37	−0.01	0.12	−0.03	−0.05
	SCS	0.01	−0.02	0.03	0.25	−0.21	−0.26	−0.05	0.00	0.04
	CS	0.01	−0.02	0.00	0.00	**0.05**	0.66	0.12	−0.17	−0.25
	HL	0.12	0.09	0.11	−0.08	0.15	**0.13**	0.37	−0.19	−0.24
	Persistency	0.05	0.00	0.01	−0.04	0.00	0.08	**0.19**	−0.05	−0.09
	Dystocia	−0.04	−0.03	−0.03	−0.01	−0.04	0.03	0.01	**0.04**	0.89
	Stillbirth	−0.03	−0.03	−0.03	−0.01	−0.03	0.00	0.01	0.36	**0.02**
2008–	Milk	**0.51**	0.45	0.85	0.19	−0.31	0.09	0.23	−0.07	0.00
2016	Fat	0.58	**0.50**	0.61	0.08	−0.26	0.09	0.08	0.00	0.04
	Protein	0.90	0.68	**0.45**	0.20	−0.38	0.06	0.12	−0.06	0.05
	SCS	0.03	−0.01	0.04	**0.25**	−0.26	−0.25	−0.05	0.05	0.05
	CS	−0.01	−0.04	−0.03	0.00	**0.05**	0.53	0.05	−0.25	−0.20
	HL	0.12	0.09	0.11	−0.06	0.14	0.13	0.40	−0.19	−0.19
	Persistency	0.05	0.01	0.01	−0.04	−0.02	0.09	**0.22**	0.00	−0.02
	Dystocia	−0.03	−0.02	−0.03	−0.01	−0.04	0.03	0.01	**0.04**	0.73
	Stillbirth	−0.03	−0.02	−0.03	−0.01	−0.02	−0.01	0.01	0.29	**0.02**

**Table 4 vetsci-09-00434-t004:** Genetic parameters for protein and conception status ± standard errors in the first 3 parities. Heritabilities on the diagonal. Genetic correlations above the diagonal and environmental correlations below the diagonal.

	Protein 1	Protein 2	Protein 3	CS 1	CS 2	CS 3
Protein 1	0.446 ± 0.006	0.842 ± 0.007	0.684 ± 0.013	−0.338 ± 0.027	−0.294 ± 0.030	−0.305 ± 0.036
Protein 2	0.392 ± 0.005	0.316 ± 0.006	0.954 ± 0.004	−0.163 ± 0.032	−0.193 ± 0.033	−0.237 ± 0.039
Protein 3	0.274 ± 0.006	0.434 ± 0.004	0.250 ± 0.007	−0.008 ± 0.034	−0.049 ± 0.037	−0.121 ± 0.043
CS 1	0.053 ± 0.004	0.059 ± 0.004	0.088 ± 0.004	0.054 ± 0.003	0.959 ± 0.011	0.895 ± 0.022
CS 2	−0.029 ± 0.005	0.044 ± 0.005	0.063 ± 0.004	0.052 ± 0.003	0.047 ± 0.003	0.992 ± 0.020
CS 3	−0.020 ± 0.006	−0.016 ± 0.004	0.061 ± 0.005	0.038 ± 0.004	0.062 ± 0.004	0.048 ± 0.004

**Table 5 vetsci-09-00434-t005:** Genetic and environmental variance components from the analysis of data set 1 for cows born from 2008–2016.

Variance Component		Milk	Fat	Protein	SCS	CS	Herd-Life	Persistency	DC	SB
Genetic	Milk	1,104,690.4	18,187.0	24,365.4	109.7	−2604.3	20,562.9	1428.5	−397.7	13.7
	Fat	18,187.0	1491.4	641.0	1.7	−78.9	789.7	17.7	−0.2	4.3
	Protein	24,365.4	641.0	752.4	3.0	−82.3	385.9	18.8	−8.8	3.9
	SCS	109.7	1.7	3.0	0.3	−1.1	−29.5	−0.2	0.1	0.1
	CS	−2604.3	−78.9	−82.3	−1.1	62.1	917.6	2.4	−10.3	−4.7
	Herd-life	20,562.9	789.7	385.9	−29.5	917.6	48,508.9	521.4	−215.0	−122.3
	Persistency	1428.5	17.7	18.8	−0.2	2.4	521.4	34.7	−0.1	−0.3
	Dystocia	−397.7	−0.2	−8.8	0.1	−10.3	−215.0	−0.1	27.0	11.2
	Stillbirth	13.7	4.3	3.9	0.1	−4.7	−122.3	−0.3	11.2	8.8
Environ-	Milk	1,072,573.5	28,444.7	29,929.6	−67.7	2086.5	89,681.7	−481.4	−767.6	−885.6
mental	Fat	28,444.7	1488.6	872.9	−2.2	2.8	2376.4	−10.0	−26.5	−26.7
	Protein	29,929.6	872.9	911.2	−1.3	43.1	2455.3	−14.3	−22.8	−26.5
	SCS	−67.7	−2.2	−1.3	0.9	1.1	−13.4	−0.4	−0.3	−0.2
	CS	2086.5	2.8	43.1	1.1	1101.3	2098.0	−10.3	−24.7	−11.8
	Herd-life	89,681.7	2376.4	2455.3	−13.4	2098.0	332,822.6	158.0	704.8	29.6
	Persistency	−481.4	−10.0	−14.3	−0.4	−10.3	158.0	123.2	4.3	3.2
	Dystocia	−767.6	−26.5	−22.8	−0.3	−24.7	704.8	4.3	669.1	149.5
	Stillbirth	−885.6	−26.7	−26.5	−0.2	−11.8	29.6	3.2	149.5	423.0

**Table 6 vetsci-09-00434-t006:** Genetic standard deviations; fraction of the Israeli breeding index, PD19; the expected and realized genetic gains (± standard errors) for cows born from 2008 through 2016.

	Genetic	Fraction	Genetic Trends		X
Trait	SD	of Index ^2^	Expected	Realized	Genetic SD
Milk (kg)	1051	0	1073.4	699.4 ± 3.5	−0.36
Fat (kg)	38.6	0.252	46.5	40.0 ± 0.1	−0.17
Protein (kg)	27.4	0.358	33.9	32.5 ± 0.1	−0.05
SCS ^1^	0.55	0.108	−0.12	−0.194 ± 0.002	0.13
CS (%)	7.9	0.135	0.3	1.91 ± 0.02	0.2
Herd-life (days)	220	0.087	152.1	209.1 ± 0.6	0.26
Persistency (%)	5.9	0.039	2.4	1.00 ± 0.02	−0.24
Dystocia (%) ^1^	5.2	0.01	−1.39	−0.800 ± 0.014	−0.11
Stillbirth (%) ^1^	3	0.012	−0.34	0.233 ± 0.009	−0.19
**PD19**	**-**	**100**	**1300.5**	**1300.7 ± 3.34**	

^1^ Negative values are economically favorable. ^2^ Computed based on equation [1].

## Data Availability

Raw data can be made available on request to J.I. Weller, subject to limitations required by the Israel Cattle Breeders’ Association.

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
