# Peer review of "Breeding Dairy Cattle for Female Fertility and Production in the Age of Genomics"

_vetsci, 2022, doi:10.3390/vetsci9080434_

Round 1
Reviewer 1 Report
Well written and presented. I have no suggestions in regards to the content. I could suggest that the title could be more specific to the content.
Author Response
No changes were required
Reviewer 2 Report
This manuscript overviews the conception status and estimated breeding values of female cattle over years. Then the authors analyzed the heritability for different traits over time. Also, the authors analyzed the genetic correlation between conception status and protein. In addition, the authors analyzed the relationships between SNPs and all traits.
The manuscript is difficult to follow for readers out of the research area. I have some comments regarding the manuscript.
Lack of names for all the Figures. The figure legends need to include all the information about the figures, but the descriptions of the current graphs are not good enough.
The definition of some indexes needs to be improved. Otherwise, the presentation of the data is a bit hard to follow.
Reviewer 3 Report
I consider the paper should undergo some revisions before being considered for publication. Firstly, the aim of the study is not presented appropriately in the abstract or in the introduction. Usually the Introduction should include a brief presentation of the topic. Tables are most likely used in the discussion section and not in the introduction.
English proofreading should be addressed.
The Discussion section does not highlight the results, the “gap of knowledge” or “new insights” that the articles brings, thus I suggest to expand on this section, considering for example studies from other countries.
Additionally, please consider other minor issues as presented below:
· Line 24: I would recommend to start the first sentence with another formulation, not with the reference “In 1960, Falconer [1]….” “Falconer [1] in 1960 noted that selection on a multiple…”
· Line 29 “. There is a extra space before “However…”
· Line 353: The following sentence is repeated in the conclusions as well. I would rather think that is more suitable for the conclusions section. “Thus, there is no apparent reason to change current 353proceduresthatcompute genomic evaluations separately for each trait.”
Round 2
Reviewer 3 Report
The current version is much better, I appreciated the extended conclusions sections.